# HM-Chromanone, a Major Homoisoflavonoid in *Portulaca oleracea* L., Improves Palmitate-Induced Insulin Resistance by Regulating Phosphorylation of IRS-1 Residues in L6 Skeletal Muscle Cells

**DOI:** 10.3390/nu14183815

**Published:** 2022-09-15

**Authors:** Jae-Eun Park, Ji-Sook Han

**Affiliations:** Department of Food Science and Nutrition, Pusan National University, Busan 46241, Korea

**Keywords:** HM-chromanone, insulin resistance, L6 skeletal muscle cells, glycogen synthesis

## Abstract

This study investigated the effect of (*E*)-5-hydroxy-7-methoxy-3-(2-hydroxybenzyl)-4-chromanone (HM-chromanone) on palmitate-induced insulin resistance and elucidated the underlying mechanism in L6 skeletal muscle cells. Glucose uptake was markedly decreased due to palmitate-induced insulin resistance in these cells; however, 10, 25, and 50 µM HM-chromanone remarkably improved glucose uptake in a concentration-dependent manner. HM-chromanone treatment downregulated protein tyrosine phosphatase 1B (PTP1B) and phosphorylation of c-Jun N-terminal kinase (JNK) and inhibitor of nuclear factor kappa-B kinase subunit beta (IKKβ), which increased because of palmitate mediating the insulin-resistance status in cells. HM-chromanone promoted insulin receptor substrate-1 (IRS-1) tyrosine phosphorylation and suppressed palmitate-induced phosphorylation of IRS-1 serine. This activated phosphoinositide 3-kinase (PI3K) and stimulated protein kinase B (AKT) phosphorylation. Phosphorylated AKT promoted the translocation of Glucose transporter type 4 to the plasma membrane and significantly enhanced glucose uptake into muscle cells. Additionally, HM-chromanone increased glycogen synthesis through phosphorylating glycogen synthase kinase 3 alpha/beta (GSK3 α/β) via AKT. Consequently, HM-chromanone may improve insulin resistance by downregulating the phosphorylation of IRS-1 serine through inhibition of negative regulators of insulin signaling and inflammation-activated protein kinases in L6 skeletal muscle cells.

## 1. Introduction

Insulin is secreted from pancreatic beta cells after food ingestion, thereby regulating blood glucose to a certain value by increasing glucose uptake into muscle cells or inhibiting the production of glucose in the liver [1]. Intracellular glucose uptake in peripheral tissues is regulated through an insulin signaling pathway and glucose transporter [2]. In particular, skeletal muscles are vital tissues in the processing of glucose, and in response to insulin, they process about 80% or more glucose [3]. However, glucose cannot be taken up into muscle cells when insulin resistance occurs, eventually raising blood glucose levels. Insulin resistance is a condition in which insulin action is lower than normal at physiological insulin concentrations [4].

Insulin resistance is a significantly dangerous constituent in the progression of diabetes, and its prevalence is rapidly increasing [5]. Obesity is a direct cause of insulin resistance development, and fat production by obesity releases free fatty acids (FFAs), causing insulin resistance. Patients with type 2 diabetes (T2D) are characterized by elevated plasma FFA levels [6]. The impairment of glucose utilization and insulin signaling has been revealed in studies with FFA treatment. Among the FFAs present in the blood, the most common FFA is palmitate, which is composed of 16-saturated carbons [7]. Insulin-resistance inducers, such as protein tyrosine phosphatase 1B (PTP1B), c-Jun N-terminal kinase (JNK), and inhibitor of nuclear factor kappa-B kinase subunit beta (IKKβ), are phosphorylated by palmitate and cause changes in the insulin receptor substrate-1 (IRS-1) residue, thereby promoting the insulin resistance indicator IRS-1 serine (IRS-1ser) [8].

The presence of IRS-1ser^307^ residue in the vicinity of the phosphotyrosine-binding (PTB) domain by an insulin-resistance inducer decreased the binding force between insulin receptor (IR) and IRS-1 and disassociated the coupling of the IRS-1 signal transduction to phosphoinositide 3-kinase (PI3K). The phosphorylation of IRS-1ser^307^ is related to decreased IRS-1-tyrosine (IRS-1tyr), phosphorylation, and insulin resistance [9,10]. As a result, increased phosphorylation of IRS-1ser^307^ and dephosphorylation of IRS-1tyr^612^ reduced glucose transporter type 4 (GLUT4)-mediated glucose uptake in muscles, resulting in insulin resistance in muscle cells [11].

In addition, defects in glycogen synthesis in muscles have been a significant factor in developing postprandial hyperglycemia and insulin resistance in patients with T2D [12,13]. Phosphorylation (inactivation) of glycogen synthase kinase 3 α/β (GSK3 α/β) by insulin leads to dephosphorylation (activation) of glycogen synthase (GS), thereby synthesizing glycogen. An obstacle in any of these elements can lead to insulin resistance [14].

*Portulaca-oleracea* L. is a Portulacaceae family that is distributed worldwide. Studies have shown that *P. oleracea* has a variety of bioactivities: antidiabetic, antibacterial, antioxidant, antiatherosclerotic, renal protection, and immune modulation [15,16,17]. (*E*)-5-hydroxy-7-methoxy-3-(2-hydroxybenzyl)-4-chromanone (HM-chromanone) from *P. oleracea* has a 16-carbon-skeleton composed of 2 phenyl rings. Our previous study of this compound revealed its efficacy in increasing glucose uptake in 3T3 adipose cells and muscle cells [15]. Despite previous studies, the mechanism by which HM-chromanone mitigates insulin resistance caused by FFAs remains unclear. Therefore, in this study, insulin resistance was induced by treating L6 skeletal muscle cells with palmitate, and the effect and molecular mechanisms of insulin resistance improvement mediated by HM-chromanone were investigated.

## 2. Materials and Methods

### 2.1. Preparation of Materials

*P. oleracea* (Taxonomic number: 20422) was purchased from Hongcheon-Hyosung-Food (Hongcheon Hyosung Food Inc., Gangwon, Hongcheon, Korea) [18]. The extraction and isolation of (*E*)-5-hydroxy-7-methoxy-3-(2-hydroxybenzyl)-4-chromanone (HMC) were conducted by a previous study [15,19]. The chemical structure and Nuclear Magnetic Resonance (NMR) of HM-chromanone are shown in Figure 1A,B. HM-chromanone: yellow gum; IR (sodium chloride, NaCl): υmax = 3400–3300, 2945, 2861, 1582, 1453, 1021 cm^−1^ UV (MeOH) λmax 204, 215, 281 nm; LREIMS m/z 300.0998 [M] + 1, mosloflavon (C_17_H_14_O_5_). HM-chromanone (2.9 mg) was harvested using powdered *P. oleracea* (300 g). It was confirmed that HM-chromanone had no toxicity, as there was no significant level up to 1000 μM (Figure 1C).

### 2.2. Preparation of Palmitate Stock Solution

The palmitate stock solution was prepared by conjugating palmitate with fatty acid-free bovine serum albumin (BSA) [20]. In brief, palmitate was dissolved in 0.1 N sodium hydroxide (NaOH) and diluted in 9.7% (*w*/*v*) of previously warmed BSA solution to yield a stock solution of 8 mM palmitate. The molar ratio of free palmitate to BSA was 6:1.

### 2.3. Cell Culture

Skeletal muscle cell line (L6, Korean cell line bank, KCLB No. 21458) was purchased from the Korean Cell Line Bank (Seoul, Korea). The cells were cultured in 90% Dulbecco’s Modified Eagle Medium (DMEM) with 10% fetal bovine serum (FBS) in 95% air and 5% carbon dioxide (CO_2_) at 37 °C. For differentiation, the cells were allowed to reach confluence, and the medium was replaced with DMEM containing 1% FBS. After differentiation, myotubes were subjected to various treatments, as described below. L6 skeletal muscle cells were treated with 0.75 mM palmitate for 16 h. The cells were then treated with a medium containing various concentrations of HM-chromanone (10, 25, and 50 µM), 5-aminoimidazole-4-carboxamide ribonucleotide (AICAR) (0.5 mM), compound C (10 µM), wortmannin (20 µM), radicicol (10 µM) or STO-609 (27 µM) for 24 h, and insulin (100 nM) for 0.5 h.

### 2.4. The 2-Deoxyglucose Uptake Assay

Insulin resistance was induced in the cells by treating them with a 0.75 mM palmitate working solution for 16 h and subsequent incubation with the following concentrations of HM-chromanone: HM-chromanone + 10 µM compound C or HM-chromanone + 20 µM wortmannin for 24 h. Glucose uptake was initiated by the addition of 10 μM 2-(N-(7-nitrobenz-2-oxa-1,3-diazol-4-yl) amino)-2 deoxyglucose (2-NBDG, Invitrogen, Carlsbad, CA, USA). Glucose uptake was measured by a fluorescence spectrophotometer (Perkin Elmer, Boston, MA, USA) set at excitation and emission wavelengths of 485 nm and 535 nm, respectively.

### 2.5. Glycogen Synthesis

Glycogen synthesis measured the conversion of D-[U-14C] glucose to glycogen in a previous study [21]. To sum it up, cells were incubated with palmitate stock solution in the wells of a 96-well for 16 h before incubating with 10, 25, and 50 µM HM-chromanone for 24 h and then incubated with insulin (100 nM) for 20 min. Cells were rapidly washed using phosphate buffered saline (PBS) and lysed by potassium hydroxide (KOH). Lysed cells were glycogen precipitated with ethanol (EtOH) for 24 h. This glycogen was dissolved in water and transferred to a scintillation vial, and the amount of cell lysate was used for protein determination.

### 2.6. Western Blotting Analysis

Cells were washed using PBS and harvested in a lysis buffer to extract total protein from the cells with insulin resistance. A sample was electrophoresed using 10% sodium dodecyl sulphate (SDS)-polyacrylamide gel. Proteins were transferred to membranes by electrophoresis in skim milk (1 h) and incubated with a primary antibody (24 h, 4 °C). Membranes were incubated with a secondary antibody for 60 min. The antigen–antibody was visualized by an enhanced chemiluminescence (ECL) reagent, and chemiluminescence was detected by a LAS-1000 (FUJIFILM, Tokyo, Japan).

### 2.7. Isolation of Plasma Membranes from L6 Skeletal Muscle Cells

The cells were treated with a medium that contained palmitate (0.75 mM) for 16h and HM-chromanone (10, 25, and 50 µM, 24h) or insulin (100 nM, 20 min). The cells were homogenized using a sonicator and centrifuged to remove non-homogenized cellular debris and nuclei from the homogenate. Another centrifugation was performed at 35,000× *g* for 1 h, and the resulting pellet was used as the plasma membrane fraction of the cells, whereas the supernatant was used as the cytosolic fraction.

### 2.8. Immunostaining and Microscopy

Cells were incubated with primary antibodies (60 min) and then incubated with secondary antibodies for 60 min. A plasma-membrane filter was prepared using sonication as described previously [22]. The background fluorescence was detected using staining membrane sheets with immunoglobulin G. The image was measured by the DeltaVision system and assayed using SoftWoRx (Applied Precision, Rača, Bratislava, Slovakia).

### 2.9. Statistical Analysis

Statistical analyses were performed using SPSS (26, IBM Corp., Armonk, NY, USA). Differences between the groups were evaluated for significance by a one-way analysis of variance followed by Duncan’s post hoc tests.

## 3. Results

### 3.1. HM-Chromanone Increases Glucose Uptake

Under insulin-stimulated conditions, compared to control cells, palmitate treatment decreased glucose uptake by 0.45-fold (Figure 1D). However, HM-chromanone treatment increased glucose uptake; glucose uptake increased by 0.51 ± 0.04, 0.81 ± 0.07, and 0.93 ± 0.05-fold upon treatment with HM-chromanone at concentrations of 10, 25, and 50 µM. Consequently, HM-chromanone under insulin-stimulated conditions may effectively increase glucose uptake in insulin-resistant cells.

### 3.2. HM-Chromanone Decreases PTP1B, JNK, and IKKβ Expression

HM-chromanone markedly inhibited the activation of PTP1B in cells with insulin resistance (Figure 2). Under insulin-stimulated conditions, cells treated with palmitate showed 316.85 ± 5.51% activation of PTP1B; but 10, 25, and 50 µM of HM-chromanone treatment resulted in significant inhibition of PTP1B, to 242.69 ± 11.64%, 194.38 ± 15.27%, and 117.97 ± 6.72%, respectively. Additionally, HM-chromanone significantly inhibited the phosphorylation of JNK and IKKβ, known as inflammation-activated protein kinases in cells with insulin resistance. Figure 2 shows that the levels of phosphorylated (p)-JNK (263.01 ± 6.81%) and p-IKKβ (259.42 ± 4.94%) significantly increased in palmitate-treated L6 skeletal muscle cells. However, 10, 25, and 50 µM of HM-chromanone significantly reduced the phosphorylation of JNK to 220.00 ± 10.00%, 170.56 ± 12.55%, and 143.39 ± 10.19%, respectively. In addition, HM-chromanone inhibited the activation of IKKβ. The phosphorylation of IKKβ was inhibited by 230.32 ± 10.33%, 187.70 ± 12.21%, and 117.21 ± 5.21% (treatment with palmitate; 259.42 ± 4.94%). Consequently, HM-chromanone under insulin-stimulated conditions might inhibit the activation of PTP1B and the phosphorylation of JNK and IKKβ.

### 3.3. HM-Chromanone Regulates IRS Residues

HM-chromanone upregulated tyrosine phosphorylation and downregulated serine phosphorylation of IRS-1 (Figure 3). The IRS-1tyr phosphorylation was 40.45 ± 7.39% in cells; however, the treatment of HM-chromanone enhanced IRS-1tyr phosphorylation to 58.97 ± 2.64%, 79.48 ± 2.75%, and 97.43 ± 6.26% (10, 25, and 50 µM, respectively), whereas p-IRS-1ser was inhibited to 231.78 ± 20.03%, 155.81 ± 5.02%, and 123.25 ± 10.18% (treatment with palmitate; 280.62 ± 9.69%), respectively.

### 3.4. HM-Chromanone Modulates the Activation of Sub-Kinases of IRS-1

The activation of PI3K and the phosphorylation of AKT and AS160 (the sub-kinase of AKT) decreased by 34.58 ± 7.82%, 42.25 ± 2.50%, and 29.08 ± 3.57%, respectively, compared to control cells (Figure 4). However, the activation of PI3K and the phosphorylation of AKT and AS160 increased because of the treatment with HM-chromanone. HM-chromanone increased the activation of PI3K by 49.24 ± 3.13%, 59.39 ± 2.73%, and 84.58 ± 3.84% (10, 25, and 50 µM, respectively). The phosphorylation of AKT increased to 50.42 ± 2.33%, 61.97 ± 5.30%, and 76.62 ± 5.09%, whereas that of AS160 increased to 36.77 ± 3.37%, 62.01 ± 4.17%, and 76.92 ± 4.55% by treatment with 10, 25, and 50 µM HM-chromanone, respectively. Consequently, HM-chromanone treatment under insulin-stimulated conditions significantly restored the activation of the sub-kinases of IRS-1 in cells with insulin resistance.

### 3.5. HM-Chromanone Upregulates the Expression of Plasma Membrane (PM)-GLUT4

The expression of plasma membrane-GLUT4 (PM-GLUT4) decreased by 35.56 ± 4.72% compared to control cells (Figure 5). However, treatment with HM-chromanone increased the expression of PM-GLUT4 in L6 skeletal muscle cells with insulin resistance. The expression of PM-GLUT4 was upregulated by 10, 25, and 50 µM of HM-chromanone to 57.39 ± 2.70%, 67.25 ± 5.17%, and 86.62 ± 8.37%, respectively.

In Figure 5C, HM-chromanone treatment, like insulin treatment, triggered L6 skeletal muscle cells to display a ring-like distribution of GLUT4, suggesting that more GLUT4 molecules are localized in the plasma membrane or beneath the plasma membrane in these cells. There was a significant reduction in the mean fluorescence intensity on the plasma membranes of palmitate-treated L6 skeletal muscle cells. In contrast, HM-chromanone (10, 25, and 50 µM) appeared markedly different from control cells. Especially, the effects of 50 µM HM-chromanone were almost comparable to the effect of insulin in recruiting GLUT4 to the plasma membrane.

### 3.6. HM-Chromanone Regulates Glycogen Synthesis Enzymes

GSK3 α/β has been identified as a target of AKT, a serine/threonine kinase located downstream of phosphatidylinositol 3-kinase [23]. Phosphorylation (inactivation) of GSK3 α/β induces dephosphorylation and activation of GS to promote glycogenesis [24]. Thus, to investigate HM-chromanone’s effects under insulin-treated conditions on glycogenesis in L6 skeletal muscle cells with insulin resistance, the levels of phosphorylated GSK-3 α/β and GS were determined. Figure 6 shows that palmitate treatment lowered the level of phosphorylated GSK-3 α/β, and this reduction was significantly reversed by treatment with HM-chromanone. The phosphorylation of GSK-3 α/β was 24.50 ± 3.88% in L6 skeletal muscle cells treated with palmitate, but treatment with 10, 25, and 50 µM HM-chromanone enhanced phosphorylation of GSK-3 α/β to 52.96 ± 1.53%, 62.84 ± 5.15%, and 87.35 ± 3.88%, respectively. In contrast, palmitate treatment elevated phosphorylated GS levels, but HM-chromanone treatment reversed this elevation. The phosphorylation of GS was 255.09 ± 7.60% in L6 skeletal muscle cells treated with palmitate, but HM-chromanone (10, 25, 50 µM) reduced the phosphorylation of GS to 230.53 ± 9.73%, 177.84 ± 6.19%, and 131.73 ± 4.16%, respectively. Treatment with HM-chromanone induced phosphorylation (inactivation) of GSK3 α/β and dephosphorylation (activation) of GS.

### 3.7. HM-Chromanone Improves Glycogen Synthesis

Glycogen synthesis was significantly reduced to 32.51 ± 3.81% in L6 skeletal muscle cells with insulin resistance, compared to the control cells (Figure 6C). However, after 10, 25, and 50 µM of HM-chromanone, glycogen synthesis was significantly increased to 39.69 ± 4.13%, 61.52 ± 3.41%, and 79.57 ± 5.90%, respectively. Treatment with HM-chromanone under insulin-stimulated conditions increased GSK-3 α/β phosphorylation, inactivating GSK-3 α/β, resulting in dephosphorylation (activation) of GS and consequently increased glycogen synthesis.

## 4. Discussion

T2D is characterized by defective insulin secretion and insulin resistance [25], which leads to postprandial and fasting hyperglycemia, dyslipidemia, and hyperinsulinemia [26]. Insulin resistance is generally characterized by elevated plasma FFA levels, and elevated plasma FFA inhibits glucose transport in insulin-dependent cells and leads to hyperglycemia [27]. Obese patients with T2D often show significantly impaired glucose disposal in skeletal muscles [28].

HM-chromanone, isolated from *P. oleracea* L, exerted an antidiabetic effect and improved insulin sensitivity in vitro [15]. However, little is known about the effects of HM-chromanone on insulin resistance by FFAs released in obesity in skeletal muscle cells. Thus, the potential improvement effect and molecular mechanisms of HM-chromanone on insulin resistance by treating L6 muscle cells with palmitate were investigated in this study.

Our results showed that HM-chromanone improved cell viability and promoted glucose uptake in insulin-resistant cells. Promoting glucose uptake into the cells is essential for alleviating high blood glucose levels, mainly because skeletal muscle cells consume more than 80% of plasma glucose. It has been reported that the promotion of glucose uptake into cells depends on the presence of hydroxyl or methoxyl groups in the compound [29]. HM-chromanone has two hydroxyl groups (C-2′ and C-5) and one methoxy group (C-7). According to a study, curcumin, a type of phytopolyphenol, significantly enhanced glucose uptake in insulin resistance-induced skeletal muscle cells. Curcumin consists of two hydroxyl groups and two methoxyl groups, and the enhanced glucose uptake was due to the presence of these functional groups [30].

The factors inducing insulin resistance were investigated to clarify the mechanism of action of HM-chromanone in promoting glucose uptake in insulin-resistant cells. When FFAs are increased in obesity, insulin resistance inducers, such as PTP1B, JNK, and IKKβ, are activated, thereby inducing insulin resistance [31]. PTP1B, a negative regulator of insulin signaling, is widely expressed in insulin-sensitive tissues and acts through dephosphorylation of IRS-1tyr residues [32]. Palmitate induces insulin resistance by enhancing PTP1B expression in insulin target tissues [33].

Additionally, the activities of both IKKβ and JNK are elevated in obesity, and these kinases are essential for the production of inflammatory mediators and in the desensitization of insulin signaling [34]. JNK activation leads to phosphorylation of IRS-1ser; this disrupts insulin signaling, leading to insulin resistance, and ultimately contributing to the pathogenesis of T2D [35]. The phosphorylation of both JNK and IKKβ increased in our study, similar to other studies on palmitate-induced insulin-resistant L6 skeletal muscle cells [36]. However, treatment with HM-chromanone inhibited the phosphorylation of JNK and IKKβ in insulin-resistance-induced L6 skeletal muscle cells. Quercetin, a flavonol from the flavonoid group of polyphenols, attenuated phosphorylation of JNK and IKKβ in insulin-resistance-induced L6 skeletal muscle cells [37].

IRS-1ser phosphorylation (i.e., ser307) leads to an insulin signaling blockade by inhibiting IRS-1tyr phosphorylation induced by insulin. In turn, this action inhibits downstream signaling pathways to promote glucose uptake. Among them, it has been reported that serine phosphorylation of IRS-1, an insulin resistance indicator induced by insulin-resistant inducers such as JNK and IKKβ, is critical to developing insulin resistance [38,39,40]. This study demonstrated that FFA elevation resulted in increased phosphorylation of IRS-1ser^307^ in L6 skeletal muscle cells. However, after HM-chromanone treatment, the phosphorylation of IRS-1ser^307^ decreased, and the phosphorylation of IRS-1tyr612 increased. The decrease in IRS-1ser^307^ phosphorylation seems to be due to the inhibition of insulin resistance inducers such as PTP1B, JNK, and IKKβ by HM-chromanone. It has also been reported that inhibition of several kinases, including JNK, IKKβ, and PTP1B, resulted in decreased phosphorylation of IRS-1ser^307^ [41,42].

Deng et al. [30] reported that flavonoids inhibited the phosphorylation of IRS-1ser^307^ and increased IRS-1tyr phosphorylation, suggesting an ameliorative effect against FFA-induced insulin resistance in C2C12 muscle cells. Polyphenol compounds have a benzene ring structure substituted with a hydroxyl group and are classified into flavonoids and nonflavonoids according to their chemical structure. Flavonoid compounds (such as flavones, flavonols, isoflavones, and homoisoflavonoids) generally have a typical structure of C_6_-C_3_-C_6_ and exist as glycosides bound through a hydroxyl group. One study reported that the presence of multiple hydroxyl groups in flavonoids tends to improve glucose uptake through the IRS pathway by modulating IRS-1ser^307^ and IRS-1tyr phosphorylation [43,44]. HM-chromanone is a subclass of flavonoids with two hydroxyl groups and has a heterocyclic C_6_-C_3_-C_6_ ring structure. Thus, it improves glucose uptake by regulating the IRS-1 residues in insulin-resistant cells due to its functional groups.

Furthermore, AKT directly phosphorylates AS160 and increases glucose uptake by stimulating the translocation of GLUT4 to the PM in skeletal muscle cells [45]. In agreement with previous research [46], the induction of insulin resistance by palmitate treatment reduced AS160 phosphorylation and reduced the PM-GLUT4 content in this study. However, HM-chromanone treatment restored AS160 phosphorylation and enhanced PM-GLUT4 expression in impaired L6 skeletal muscle cells by palmitate.

Impaired glycogen synthesis and decreased glucose uptake have been found in insulin-resistant T2D patients [47]. GSK-3 α/β, an essential enzyme for glycogen synthesis, is a substrate of the AKT signaling pathway, and phosphorylation (activation) of GSK-3 α/β is regulated by AKT-mediated phosphorylation [48]. In this study, treatment with HM-chromanone increased GSK-3 α/β phosphorylation, which resulted in the activation of GS and consequently increased glycogen synthesis. Kaempferol 3-neohesperidoside, the major flavonoid found in *Bauhinia forficata* leaves, stimulated glycogen synthesis in rat muscle, and its stimulatory effect on glycogen synthesis was due to the activation of AKT-GSK3 [49]. Kaempferol has shown that biological activities are highly dependent on the various functional groups attached to the phenolic groups [50]. HM-chromanone has a hydroxyl group in C-1′ and C-2′, corresponding to the B-ring, and like kaempferol, it has a double bond between C-4 and C-5 and a hydroxyl group in C-5. Therefore, we suppose that HM-chromanone increases glycogen synthesis due to this structural feature.

## 5. Conclusions

In conclusion, HM-chromanone decreased PTP1B, JNK, and IKKβ and then downregulated the phosphorylation of IRS-1ser and upregulated the phosphorylation of IRS-1tyr. Furthermore, HM-chromanone restored PI3K/AKT pathway activation, and glucose uptake was impaired by palmitate. Additionally, the activation of AKT by HM-chromanone led to the phosphorylation of GSK3 α/β and subsequent activation of GS, resulting in an increase in glycogen synthesis. Taken together, HM-chromanone improved palmitate-induced insulin resistance by decreasing insulin resistance inducers and regulating phosphorylation of IRS-1 residues in L6 skeletal muscle cells (Figure 7). HM-chromanone has the potential to be a glucose homeostasis regulating agent and elucidates the potential antidiabetic effect of a substance isolated from natural products. Further research should be conducted on several aspects. In addition, if HM-chromanone is used clinically for medical purposes, a special permit procedure is required, and research on intake as a functional food should be conducted in the future in vivo.

## Figures and Tables

**Figure 1 nutrients-14-03815-f001:**
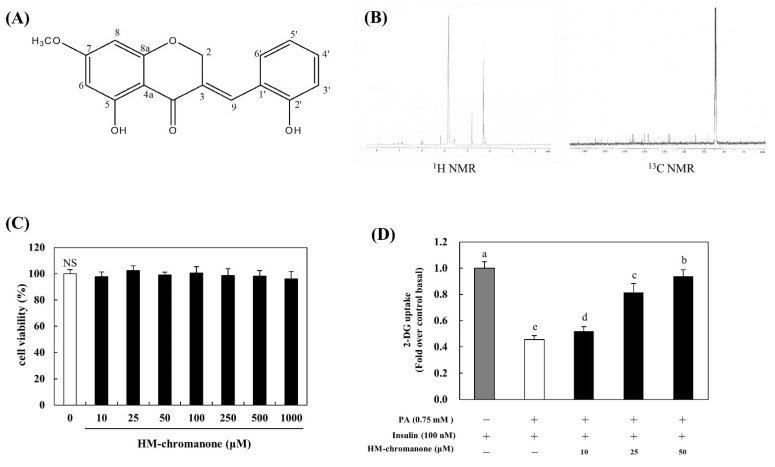
Chemical structure, NMR, and glucose uptake of HM-chromanone. (**A**) Chemical structure of HM-chromanone. (**B**) ^1^H NMR and ^13^C NMR spectra. (**C**) Cell viability of HM-chromanone. (**D**) Glucose uptake effect of HM-chromanone. L6 skeletal muscle cells were treated with 0.75 mM palmitate (PA) for 16 h. Thereafter, the cells were treated with HM-chromanone at concentrations of 10, 25, and 50 μM for 24 h in the presence of insulin (100 nM) before the 2-deoxyglucose uptake assay. Insulin-treated cell culture (100 nM) was used as a positive control. Each value is expressed as the mean ± standard deviation (*n* = 3), and values with different superscript letters are significantly different with *p* < 0.05, as determined using Duncan’s multiple range tests. PA: palmitate; NMR: Nuclear Magnetic Resonance; HM-chromanone: (*E*)-5-hydroxy-7-methoxy-3-(2-hydroxybenzyl)-4-chromanone; ^1^H: hydrogen; ^13^C: carbon-13; NS: not significant; 2-DG: 2-deoxyglucose.

**Figure 2 nutrients-14-03815-f002:**
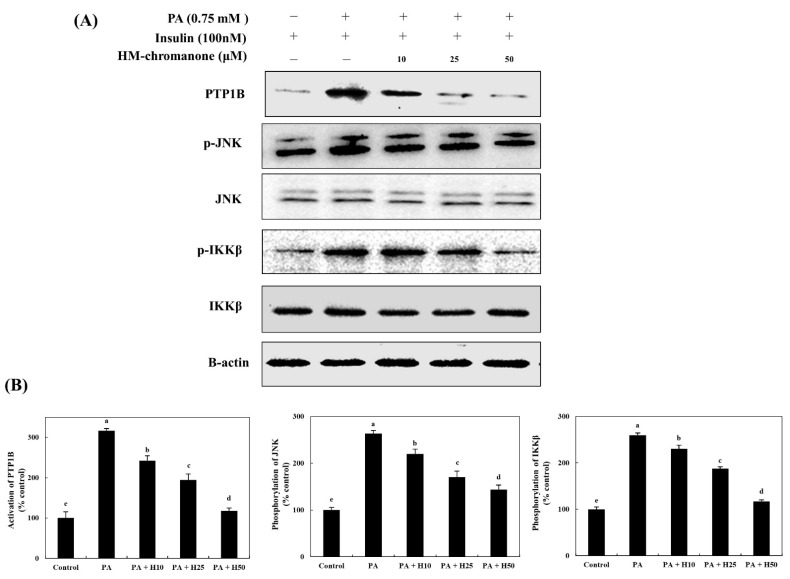
HM-chromanone decreases PTP1B, JNK, and IKKβ expression. Cells were treated with 0.75 mM palmitate (PA) for 16 h, and the cells were treated with HM-chromanone at concentrations of 10, 25, and 50 μM or insulin (100 nM) for 24 h in the presence of insulin (100 nM). (**A**) Phosphorylated PTP1B, JNK, and IKKβ levels. (**B**) Expression levels of PTP1B, JNK, and IKKβ. Each value is expressed as the mean ± SD (*n* = 3), and values with different superscript letters are significantly different at *p* < 0.05, as determined by using Duncan’s tests. PA: palmitate; PTP1B: protein tyrosine phosphatase 1B; JNK: c-Jun N-terminal kinase; p-JNK: phospho-JNK; IKKβ: inhibitor of nuclear factor kappa-B kinase subunit beta; p-IKKβ: phospho-IKKβ; SD, standard deviation; H10: HM-chromanone 10 μM; H25: HM-chromanone 25 μM; H50: HM-chromanone 50 μM.

**Figure 3 nutrients-14-03815-f003:**
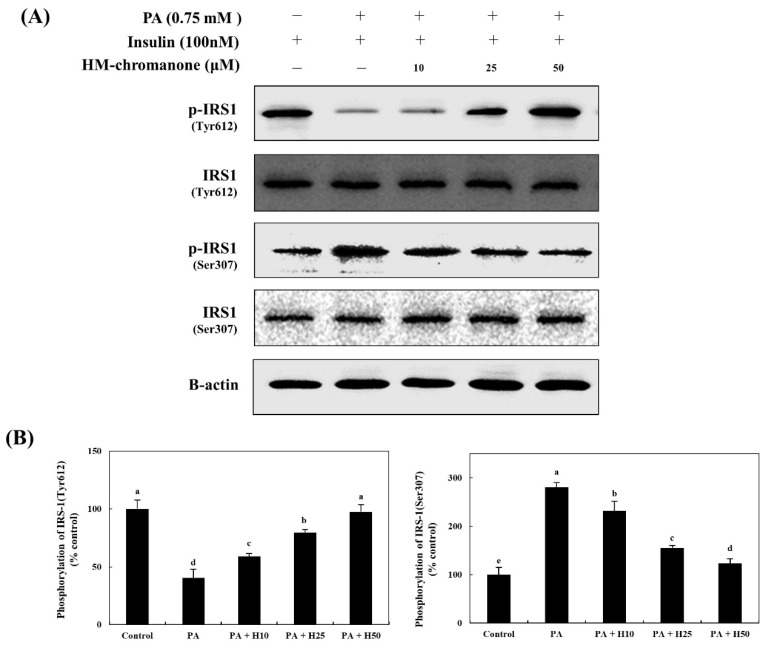
HM-chromanone increases tyrosine phosphorylation and decreases serine phosphorylation of IRS-1. Cells were treated with 0.75 mM palmitate (PA) for 16 hand with HM-chromanone at concentrations of 10, 25, and 50 μM for 24 h in the presence of insulin (100 nM). (**A**) Phosphorylated IRS-1 (tyr612) and IRS-1 (ser307) levels. (**B**) Expression levels of IRS-1 (tyr612) and IRS-1 (ser307). Each value is expressed as the mean ± SD (*n* = 3), and values with different superscript letters are significantly different at *p* < 0.05, as determined by using Duncan’s tests. PA: palmitate; IRS1: insulin receptor substrate-1; p-IRS1: phospho-IRS1; H10: HM-chromanone 10 μM; H25: HM-chromanone 25 μM; H50: HM-chromanone 50 μM.

**Figure 4 nutrients-14-03815-f004:**
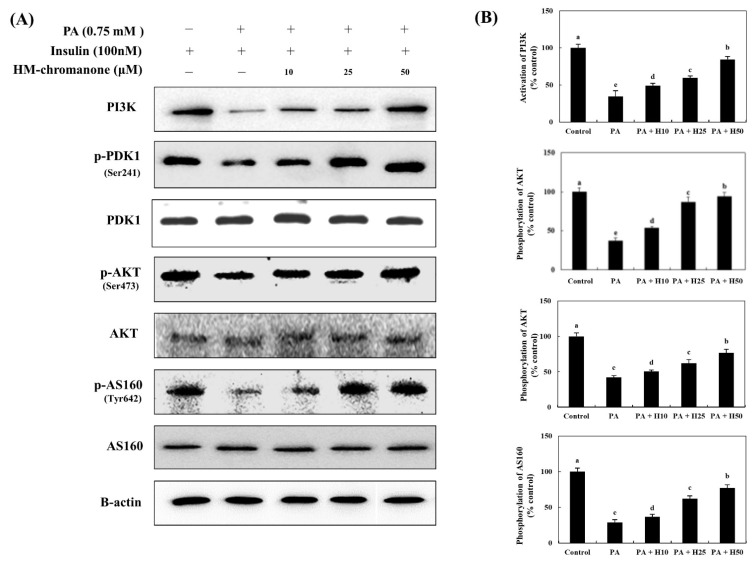
HM-chromanone modulates the activation of sub-kinases of IRS-1. Cells were treated with 0.75 mM palmitate (PA) for 16 h and with HM-chromanone at concentrations of 10, 25, and 50 μM for 24 h in the presence of insulin (100 nM). (**A**) Phosphorylation of PI3K, PDK, AKT, and AS160 and their corresponding expression levels. (**B**) Expression levels of PI3K, PDK, AKT, and AS160. Each value is expressed as the mean ± SD (*n* = 3), and values with different superscript letters are significantly different at *p* < 0.05, as determined by using Duncan’s tests. PA: palmitate; PI3K: phosphoinositide 3-kinase; PDK: phosphoinositide-dependent protein kinase; p-PDK1: phospho-phosphoinositide-dependent protein kinase-1; AKT: protein kinase B; p-AKT: phospho-protein kinase B; H10: HM-chromanone 10 μM; H25: HM-chromanone 25 μM; H50: HM-chromanone 50 μM.

**Figure 5 nutrients-14-03815-f005:**
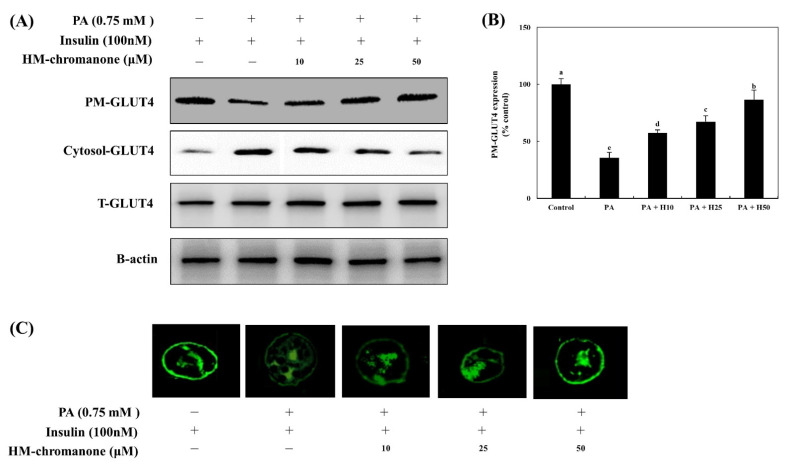
HM-chromanone upregulates the expression of plasma membrane-GLUT4. Cells were treated with 0.75 mM palmitate (PA) for 16 h and with HM-chromanone at concentrations of 10, 25, and 50 μM for 24 h in the presence of insulin (100 nM). (**A**) Phosphorylation levels of PM- or cytosolic GLUT4. (**B**) Expression levels of PM- or cytosolic GLUT4. (**C**) Immunofluorescence analysis for GLUT4 expression on PM. Each value is expressed as the mean ± SD (*n* = 3), and values with different superscript letters are significantly different at *p* < 0.05, as determined by using Duncan’s tests. PA: palmitate; GLUT-4: glucose transporter type 4; PM-GLUT4: plasma membrane-GLUT4; H10: HM-chromanone 10 μM; H25: HM-chromanone 25 μM; H50: HM-chromanone 50 μM.

**Figure 6 nutrients-14-03815-f006:**
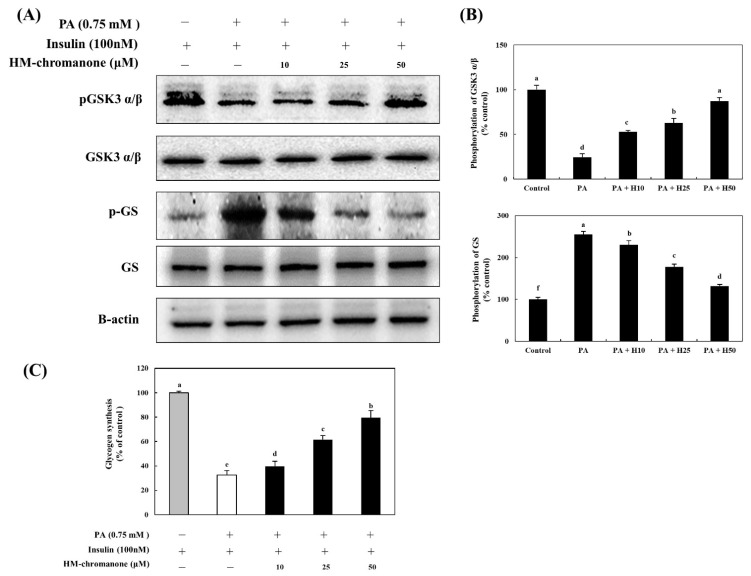
HM-chromanone regulates the activation of GSK α/β and GS and improves the glycogen synthesis. Cells were treated with 0.75 mM palmitate (PA) for 16 h and with HM-chromanone at concentrations of 10, 25, and 50 μM for 24 h in the presence of insulin (100 nM). (**A**) Phosphorylation levels of GSK3 α/β and GS. (**B**) Expression levels of GSK3 α/β and GS. (**C**) Effect of HM-chromanone on glycogen synthesis. Expression levels of PM- or cytosolic GLUT4. Each value is expressed as the mean ± SD (*n* = 3), and values with different superscript letters are significantly different at *p* < 0.05, as determined by using Duncan’s tests. PA: palmitate; GSK α/β: glycogen synthase kinases 3 alpha/beta; pGSK α/β: phospho-GSK α/β; GS: glycogen synthase; p-GS: phospho-glycogen synthase; H10: HM-chromanone 10 μM; H25: HM-chromanone 25 μM; H50: HM-chromanone 50 μM.

**Figure 7 nutrients-14-03815-f007:**
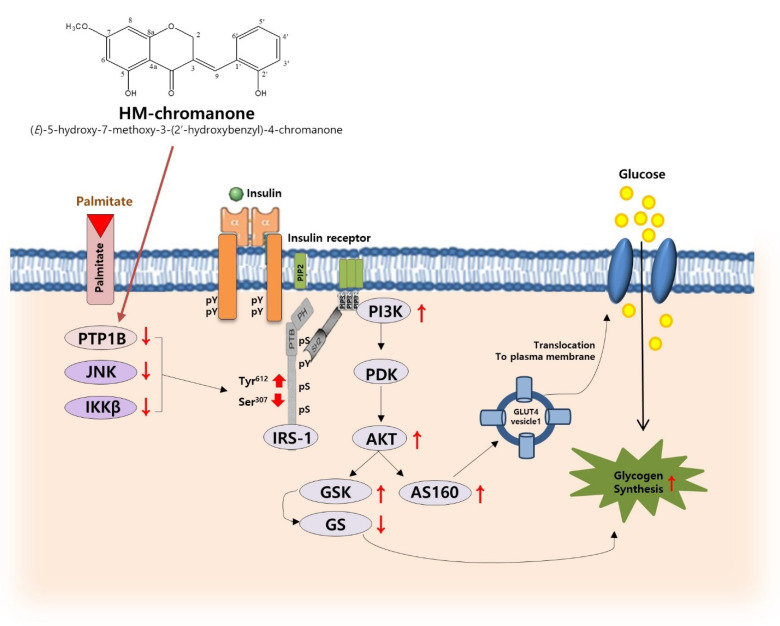
Proposed mechanism for improvement of insulin resistance by regulating phosphorylation of IRS-1 residues and glycogen synthesis in L6 skeletal muscle cells with palmitate-induced insulin resistance. pY: phosphotyrosine; PIP2: Phosphatidylinositol 4,5-bisphosphate; PIP3: Phosphatidylinositol (3,4,5)-trisphosphate; SH2: Src homology 2; PTB: phosphotyrosine binding; PH: pleckstrin homology; pS: phosphoserine. The red arrows mean increase or decrease.

## Data Availability

Not applicable.

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
