# Peer review of "HM-Chromanone, a Major Homoisoflavonoid in Portulaca oleracea L., Improves Palmitate-Induced Insulin Resistance by Regulating Phosphorylation of IRS-1 Residues in L6 Skeletal Muscle Cells"

_nutrients, 2022, doi:10.3390/nu14183815_

Round 1
Reviewer 1 Report
This research manuscript “HM-Chromanone, a Major Homoisoflavonoid in Portulaca oleracea L., Improves Palmitate-Induced Insulin Resistance by Regulating Phosphorylation of IRS-1 Residues in L6 Skeletal Muscle Cells” is novel, well written and well organized. This manuscript can be accepted in its current form, and I recommend the publication of the manuscript. Few comments are needed to be addressed before it can be accepted.
1. There are numerous grammatical and typographical errors.
2. English language must be re-checked.
3. Include the details and biological activity of Portulaca oleracea L.
4. Glycogen synthesis protocol must be explained clearly.
5. Morphological and cellular changes in L6 cells pre and post treatment with various compounds must be included in the results.
6. Recent references must be included.
7. Future direction of the study must be included.
Author Response
Author's Reply to the Review Report (Reviewer 1)
This research manuscript “HM-Chromanone, a Major Homoisoflavonoid in Portulaca oleracea L., Improves Palmitate-Induced Insulin Resistance by Regulating Phosphorylation of IRS-1 Residues in L6 Skeletal Muscle Cells” is novel, well written and well organized. This manuscript can be accepted in its current form, and I recommend the publication of the manuscript. Few comments are needed to be addressed before it can be accepted.
- There are numerous grammatical and typographical errors.
- English language must be re-checked.
→ We grammatically re-edited this manuscript by the native speaker "Editage".
- Include the details and biological activity of Portulaca oleracea L.
→ Portulaca oleracea L. is a perennial plant belonging to Portulacaceae and is widely distributed from temperate to tropical regions worldwide. Recent studies reported that P. oleracea extract has various bioactivities, including antidiabetes, antimicrobial, antioxidant, anti-atherogenic, renoprotective and immunomodulatory activities. (E)-5-hydroxy-7-methoxy-3-(2-hydroxybenzyl)-4-chromanone (HM-chromanone) from P. oleracea was isolated and had 16-carbon skeleton structure consisting of two phenyl rings (A and B) and heterocyclic ring (C).
- Glycogen synthesis protocol must be explained clearly.
→ Glycogen synthesis was assessed by measuring the incorporation of D-[U-14C] glucose into glycogen as described previously [21]. Briefly, following 4 h of serum starvation, cells (1 × 104 cells/well) were pre-incubated with palmitate stock solution in the wells of a 96-well plate for 16 h before incubating with 10, 25, and 50 µM HM-chromanone for 24 h and then incubated with insulin (100 nM) for 20 min. Cells were then quickly washed in ice-cold PBS and lysed in 0.2 ml of KOH (1 mol/l). Cell lysates were glycogen precipitated with ethanol for overnight. Precipitated glycogen was dissolved in water and transferred to scintillation vials for radioactivity counting. An aliquot of the cell lysates was used for protein determination.
◆ Reference
Ceddia R, Sweeney G (2004) Creatine increases glucose oxidation and AMPK phosphorylation and reduces lactate production in L6 rat skeletal muscle cells. J Physiol 555:409–421
- Morphological and cellular changes in L6 cells pre and post treatment with various compounds must be included in the results.
→ We included in the result on morphological and cellular changes in L6 cells pre and post treatment with various compounds.
There was a significant reduction in the mean fluorescence intensity at the plasma membranes of palmitate-treated L6 skeletal muscle cells. In contrast, palmitate-treated L6 skeletal muscle cells with HM-chromanone (10, 25 and 50 µM) appeared markedly different from control cells. Especially, these effects of 50 µM HM-chromanone were almost comparable to the effect of insulin in recruiting GLUT4 to the plasma membrane.
- Recent references must be included.
→ Cited articles have been replaced with recently published papers.
- Future direction of the study must be included.
→ HM-chromanone decreased PTP1B, JNK, and IKKβ, which are insu-lin resistance inducers induced by palmitate, and then downregulated the phosphorylation of IRS-1ser and upregulated the phosphorylation of IRS-1tyr in palmitate-induced insulin-resistant L6 skeletal muscle cells. Furthermore, HM-chromanone restored the PI3K/AKT pathway activation, and glucose uptake was impaired by palmitate. Additionally, the activation of AKT by HM-chromanone led to the phosphory-lation of GSK3 α/β and subsequent activation of GS, resulting in an increase in glycogen synthesis in insulin-resistant cells. These results collectively indicate that HM-chromanone improved palmitate-induced insulin resistance by decreasing insulin resistance inducers and regulating phosphorylation of IRS-1 residues in L6 skeletal muscle cells (Figure 7). HM-chromanone has potential as a glucose homeostasis regulating agent and elucidate the potential antidiabetic effect of substance isolated from natural products, research should be conducted from in several aspects. In addition, if HM-chromanone is used clinically for medical purposes, a special permit procedure is required, and research on intake as a functional food should be conducted in the future in in vivo.
Reviewer 2 Report
Dear Authors,
Congratulations on writing such an interesting article. This manuscript aims to reveal the effect and molecular mechanisms of insulin resistance (IR) improvement mediated by HM-chromanone in skeletal muscle cell line L6 where IR is induced through palmitate treatment.
The following are my comments and suggestions:
1. Kindly provide the information about the HM-chromanone quantification and structural identification in P. oleracea, extract in brief under the method section.
2. Does any observation notice in absence of insulin stimulation? Please describe it in detail in respect of the molecular pathways with and without PA stimulation.
3. I was wondering why the author did not use or show classical inflammatory markers such as IL-6, TNF-b, IL-1a, etc. Kindly provide information about them and other cytokines and how they get modulated during the IR state.
Discussion:
Kindly provide brief details about the metabolism of HM- HM-chromanone and how they get modulated or affects in case of Diabetic or IR conditions. Kindly explain the secondary metabolites originating from or via HM -chromanone and their potential role in IR.
What is the most interesting feature that makes HM-chromanone a potential therapeutic agent? Kindly use a few bioinformatics tools and please predict the GO and KEGG pathways that get affected by HM-chromanone.
Just a suggestion, kindly replace the references which are older than 10 years with recently published articles unless cited articles are the best in their field.

Author Response
Author's Reply to the Review Report (Reviewer 2)
Congratulations on writing such an interesting article. This manuscript aims to reveal the effect and molecular mechanisms of insulin resistance (IR) improvement mediated by HM-chromanone in skeletal muscle cell line L6 where IR is induced through palmitate treatment.
The following are my comments and suggestions:
- Kindly provide the information about the HM-chromanone quantification and structural identification in P. oleracea, extract in brief under the method section.
→ The chemical structure and NMR of HM-chromanone are shown in Figure 1(A, B). HM-chromanone: yellow gum; IR (NaCl): υmax=3400-3300, 2945, 2861, 1582, 1453, 1021 cm-1 UV (MeOH) λmax 204, 215, 281 nm; LREIMS m/z 300.0998 [M]+1, C17H14O5. Dried P. oleracea powder (300 g) was extracted to obtain 2.9 mg of HM-chromanone.
- Does any observation notice in absence of insulin stimulation? Please describe it in detail in respect of the molecular pathways with and without PA stimulation.
→ In this study, basal control is an insulin stimulated condition, and another control is PA+insulin treatment. Insulin (100 nM) was used as a positive control to compare with the test sample in palmitate-induced insulin-resistant L6 skeletal muscle cells. The cells were treated with a medium containing insulin (100 nM) for 20 min. According to the study, 100 nM insulin treatment within 30 min stimulates the canonical IRS-PI3K-Akt pathway under physiological conditions, promotes GLUT4 translocation to the membrane from inner vesicules and consequently stimulates glucose uptake.
The following is references using the control (Ins, Ins+PA) as in this study.
◆ References
① Yong Zhou et al., Ampelopsin Improves Insulin Resistance by Activating PPARγ and Subsequently Up-Regulating FGF21-AMPK Signaling Pathway. PLOS ONE. July 8, 2016
② Li HB, Yang YR, Mo ZJ, Ding Y, Jiang WJ. Silibinin improves palmitate-induced insulin resistance in C2C12 myotubes by attenuating IRS-1/PI3K/Akt pathway inhibition. Braz J Med Biol Res. 2015;48(5):440-446.
→ Insulin binds to the insulin receptor (IR) and initiates its action by inducing tyrosine phosphorylation of insulin receptor substrate -1 (IRS-1). Then, phosphatidyl inositol-3 kinase (PI3K) and protein kinase B (Akt) are activated, stimulating the translocation of glucose transporter 4 (GLUT4) to the plasma membrane, resulting in increased glucose uptake. Elevated levels of plasma free fatty acids (FFA) promote serine phosphorylation of IRS-1 by activating/phosphorylating protein kinases, thereby interfering with insulin signaling. Phosphorylated IRS-1Ser is known to negate insulin receptor tyrosine kinase signaling, which leads to insulin resistance.
- I was wondering why the author did not use or show classical inflammatory markers such as IL-6, TNF-b, IL-1a, etc. Kindly provide information about them and other cytokines and how they get modulated during the IR state.
→ We investigated inflammatory markers such as IL-6, TNF-a, IL-1b, etc. in follow-up study. The objective of first follow-up study was to identify the anti-inflammatory effects of HM-chromanone in lipopolysaccharide (LPS)-stimulated RAW 264.7 macrophages. HM-chromanone was observed to significantly suppress LPS-induced expressions of inflammatory mediators such as cyclooxygenase-2 and nuclear factor -kappa B subunit 1; and inflammatory cytokines including tumor necrosis factor-α, interleukin-1β, and interleukin-6. Overall, our results suggested that HM-chromanone suppresses LPS-induced inflammation in RAW 264.7 macrophages by downregulating the expression of inflammatory factors.
At second follow-up study, we investigated the ability of HM-chromanone to attenuate the activation of inflammatory cytokines and signaling pathways associated with tumor necrosis factor (TNF)-α-mediated inflammation and insulin resistance in 3T3-L1 adipocytes. HM-chromanone improved TNF-α-mediated inflammation and insulin resistance by regulating JNK activation and the NF-κB pathway, thereby reducing inflammatory cytokine secretion and inhibiting serine phosphorylation of IRS-1 in the insulin signaling pathway in 3T3-L1 adipocytes.
→ Obesity is a direct cause of the development of insulin resistance, and fat production by obesity causes inflammation by releasing free fatty acids (FFAs). Prolonged inflammation has been reported to trigger many diseases such as diabetes and insulin resistance. Impairment of glucose utilization and insulin signaling has been ob-served in experimental studies with high-concentration FFA treatment. High-concentration FFA has been observed to result in the upregulation of proinflammatory cytokines—such as tumor necrosis factor -α (TNF-α), interleukin (IL) -1β and IL-6—and inflammatory mediators, such as inducible nitric oxide synthase (iNOS) and cyclooxygenase-2 (COX-2).
◆ References
① Kang E, Park JE, Seo Y, Han JS. (E)-5-hydroxy-7-methoxy-3-(2'-hydroxybenzyl)-4-chromanone isolated from Portulaca oleracea L. suppresses LPS-induced inflammation in RAW 264.7 macrophages by downregulating inflammatory factors. Immunopharmacol Immunotoxicol. 2021;43(5):611-621.
② Park JE, Kang E, Han JS. HM-chromanone attenuates TNF-α-mediated inflammation and insulin resistance by controlling JNK activation and NF-κB pathway in 3T3-L1 adipocytes. Eur J Pharmacol. 2022;921:174884.
Discussion:
- Kindly provide brief details about the metabolism of HM-chromanone and how they get modulated or affects in case of Diabetic or IR conditions. Kindly explain the secondary metabolites originating from or via HM -chromanone and their potential role in IR.
→ HM-chromanone decreased PTP1B, JNK, and IKKβ, which are insulin resistance inducers induced by palmitate, and then downregulated the phosphorylation of IRS-1ser and upregulated the phosphorylation of IRS-1tyr in palmitate-induced insulin-resistant L6 skeletal muscle cells. Furthermore, HM-chromanone restored the PI3K/AKT pathway activation. These results collectively indicate that HM-chromanone improved palmitate-induced insulin resistance by decreasing insulin resistance inducers and regulating phosphorylation of IRS-1 residues in L6 skeletal muscle cells.
- What is the most interesting feature that makes HM-chromanone a potential therapeutic agent? Kindly use a few bioinformatics tools and please predict the GO and KEGG pathways that get affected by HM-chromanone.
→ Previous research on HM-chromanone showed the effects of increasing glucose uptake in 3T3-L1 adipocytes and skeletal muscle cells. In this study, insulin resistance was induced by treating L6 skeletal muscle cells with palmitate, and the effect and molecular mechanisms of insu-lin resistance improvement mediated by HM-chromanone were investigated. HM-chromanone decreased insulin resistance inducers induced by palmitate, and then restored the PI3K/AKT pathway activation. Additionally, the activation of AKT by HM-chromanone led to the phosphorylation of GSK3 α/β and subsequent activation of GS, resulting in an increase in glycogen synthesis in insulin resistant cells.
→ According to the Kyoto Encyclopedia of Genes and Genomes (KEGG) pathway database, when fat production by obesity releases free fatty acids (FFA) and induces insulin resistance in muscle cells, glycogen production is reduced through TNF and insulin signaling pathways, and GLUT4 translocation and glucose uptake are inhibited. However, HM-chromanone decreased insulin resistance inducers induced by palmitate, and then restored the PI3K/AKT pathway activation. Additionally, the activation of AKT by HM-chromanone led to the phosphorylation of GSK3 α/β and subsequent activation of GS, resulting in an increase in glycogen synthesis in insulin resistant cells. Therefore, it can be predicted that HM-chromanone is affected by the KEGG pathway.
- Just a suggestion, kindly replace the references which are older than 10 years with recently published articles unless cited articles are the best in their field.
→ Cited articles have been replaced with recently published papers.